# The Disparities in Mental Health Between Gay and Bisexual Men Following Positive HIV Diagnosis in China: A One-Year Follow-Up Study

**DOI:** 10.3390/ijerph17103414

**Published:** 2020-05-14

**Authors:** Rui Luo, Vincent M.B. Silenzio, Yunxiang Huang, Xi Chen, Dan Luo

**Affiliations:** 1Department of Social Medicine and Health Management, Xiangya School of Public Health, Central South University, Changsha 410078, Hunan, China; 186911059@csu.edu.cn (R.L.); waterhux@163.com (Y.H.); 2Department of Urban-Global Public Health, Rutgers School of Public Health, Rutgers University, Newark, NJ 07102, USA; vincent.silenzio@rutgers.edu; 3Hunan Provincial Center for Disease Prevention and Control, Changsha 410078, Hunan, China; chenxi161@sohu.com

**Keywords:** bisexual, gay, HIV/AIDS, depression, anxiety, longitudinal study

## Abstract

This study aimed to determine the change in mental health (depression and anxiety) among HIV-positive gay and bisexual men (GBM) one year after diagnosis and the disparities in trajectories of mental health between them. The potential factors contributing to the disparities were also investigated. This was a one-year follow-up study focusing on the mental health of newly diagnosed HIV-positive individuals. Participants rated their depression, anxiety, stress, and social support levels at baseline and one year later. Information on the utilization of mental healthcare and the initiation of antiretroviral therapy (ART) after diagnosis was collected at one-year follow-up. A total of 171 and 87 HIV-positive gay and bisexual men, respectively, completed two-time points surveys in this study. The depressive and anxiety symptoms experienced by HIV-positive GBM improvement one year after diagnosis. These improvements tended to be smaller in gay participants. Other factors including mental health care utilization and ART status during the one-year follow-up period, changes in social stress scores and objective social support scores were also associated with the changes in depression and anxiety, and all these factors, except for change in objective support, were found to be statistically different between HIV-positive GBM. Special attention should be given to the mental health of HIV-positive gay men. Promoting HIV-positive gay men to assess to mental health services and ART may be important for these populations to improve mental health. Enhancing social support and reducing stress levels may also be necessary for the vulnerable HIV-positive sexual minority groups.

## 1. Introduction

The mental health of HIV-positive individuals is of great concern in public health. There is evidence that HIV-positive individuals experience much higher mental health burden than the general population, with depression and anxiety being the most commonly reported [1,2]. Previous studies have found that depression and anxiety are associated with poor adherence to antiretroviral therapy (ART) and increased HIV-related mortality [3,4]. It is important to focus on the mental health among individuals living with HIV.

Due to the growing prevalence of HIV infection among them [5], the mental health of men who have sex with men (MSM) has become a topic of increasing interest. Multiple studies have summarized that MSM with HIV infection are more vulnerable to experience mental health issues, particularly depression and anxiety, compared to the general population [6,7,8]. Moreover, studies have also shown that HIV-positive MSM have worse mental health than other HIV-positive populations [9,10], while some research found no difference [11]. MSM are defined as sexual minority men from the behavioral dimension of sexual orientation, of which most self-identity as gay or bisexual men (GBM) [12]. Sexual minority men, broadly defined, (a) are men who identify themselves as gay or bisexual, (b) have sexual partners who are male, or (c) are strongly attracted to men [12]. This reflects the three dimensions of sexual orientation: identity, behavior, and attraction [13].

According to the minority stress theory, GBM are vulnerable social groups susceptible to stigma and prejudice, making them more likely to suffer from elevated stress levels [14]. A meta-analysis found that GBM have more frequent mental health problems than heterosexual men, due to minority stress [14]. In addition, although same-sex marriage is now legal in some countries around the world, it is still illegal in China. Previous studies have shown that GBM are considered losers and shamed if they do not get married and have children in China. This cultural context may make them face continued stress of being considered unfilial sons in China [15]. This excessive stress may further lead to negative mental health disorders. 

Some researchers have noticed the heterogeneity in mental health of GBM, which means that gay men and bisexual men may have disparities in mental health [16,17,18,19]. A meta-analysis has found that the depression and anxiety among bisexual men is more serious than gay men [20]. For HIV-positive GBM, is there also such a disparity between the two groups? There is a lack of research on the disparities of mental health between HIV-positive GBM. Among HIV-positive GBM, few studies have investigated the prevalence of depressive and anxiety symptoms with cross-sectional designs [21,22]. However, depressive and anxiety symptoms are time-dependent variables that may change overtime, especially in the first year after diagnosis [23]. How does the mental health of these two groups change, and are there disparities in the trajectories of mental health between them one year after being diagnosed with HIV? What factors would contribute to the disparities in trajectories of mental health between them? A detailed exploration of these issues might be valuable to develop targeted intervention to improve mental health among HIV-positive GBM.

The purposes of this study were: (1) to determine the changes in mental health (depression and anxiety) one year after HIV diagnosis and the disparities in trajectories of mental health between HIV-positive gay and bisexual men; (2) to identify the potential factors contributing to the disparities in mental health between them. According to the findings in previous studies focusing on the mental health of general GBM [20], we hypothesize that the depressive and anxiety symptoms would improve among HIV-positive GBM one year after diagnosis, and these improvements would be greater in gay participants. 

## 2. Methods 

### 2.1. Participants 

The data reported in this study derives from a longitudinal study focusing on the mental health among newly diagnosed people living with HIV. A detailed description of study design is available elsewhere [24]. Briefly, participants were consecutively recruited from the Changsha Center for Disease Control and Prevention (CDC) from March 1, 2013, to August 30, 2014. Eligible participants had to meet the following inclusion criteria: (1) at least 18 years of age; (2) receiving HIV diagnosis no more than 1 month; (3) living in Changsha for at least 6 months. 

There were 1267 people in Changsha diagnosed with HIV between 1 March, 2013, and 30 August, 2014. Of them, 855 individuals met the criteria and 557 participated in this study. Among the individuals who participated in this study, a total of 354 participants identified their sexual orientation as gay or bisexual, which was the final sample included in the analyses of this study. 

### 2.2. Measures

#### 2.2.1. Socio-Demographic Information

Socio-demographic information, including age (18–29, >29), educational background (senior or lower, college or higher), monthly income (≤4000, >4000 Yuan), marital status (single, married, divorced), employment (employed, unemployed), living alone or not, and having children or not were collected. 

#### 2.2.2. Clinical Information

Clinical information included HIV-related symptoms, CD4 counts, and ART initiation. HIV-related symptoms included persistent diarrhea, persistent cough, persistent fevers, oral thrush, unintentional weight loss, recurrent herpes simplex, and active tuberculosis. Any other symptoms were recorded. The therapeutic information (i.e., whether they initiated ART during the follow-up period) was also collected. 

#### 2.2.3. Sexual Orientation 

In this study, sexual orientation was determined by self-identification [12]. The self-identity sexual orientation of this study was measured by the question: “What is your sexual orientation?” The response options were “heterosexual”, “bisexual”, “gay”, and “not sure”.

#### 2.2.4. HIV/AIDS-Related Stress 

We used the 17 items Chinese version of the HIV/AIDS Stress Scale (CSS–HIV) to assess HIV/AIDS-related stress. It was originally developed by Pakenham [25], and then translated into Chinese by Niu and Qiu [26]. This scale has three dimensions: social stress, emotional stress, and instrumental stress. It is a 5-point Likert-type scale, with each item ranging from 0 (not at all) to 4 (extremely). Higher scores indicate a higher level of stress. The CSS–HIV has a good validity and reliability, with an overall Cronbach’s *α* coefficient of 0.906 [26].

#### 2.2.5. Social Support 

We use the 10 items Social Support Rating Scale (SSRS) developed by Xiao to evaluate the social support [27]. This scale has three dimensions: subjective support, objective support, and social support utilization, with higher scores indicating higher levels of social support. This scale has been widely used in different populations in China, including HIV-positive individuals [28,29]. The value of Cronbach’s α was 0.82 in this study.

#### 2.2.6. Mental Health Care Utilization

A single yes/no item was used to collect the mental health care utilization among participants: “Have you accessed to mental health care due to the emotional distress after diagnosis?”

#### 2.2.7. Depression 

The 9-item Patient Health Questionnaire Depression Scale (PHQ-9) was used to assess depressive symptoms. It is a 4-point Likert-type scale, with each item ranging from 0 (not at all) to 3 (almost every day) [30]. Higher scores indicate more severe depressive symptoms. A score of 10 is usually the cut-off score for significant depressive symptoms [31,32]. The Chinese version of the PHQ-9 shows good reliability and validity with a Cronbach’s *α* coefficient of 0.86 [33]. 

#### 2.2.8. Anxiety

The 7-items Generalized Anxiety Disorder Scale (GAD-7) was used to measure anxiety symptoms [34]. It is a 4-point Likert-type scale, with each item ranging from 0 (not at all) to 3 (almost every day). Higher scores indicate more severe anxiety symptoms. Similar to PHQ-9 scale, a score of 10 is usually the cut-off score for significant anxiety symptoms [35]. The Chinese version of the GAD-7 scale has been shown to have good reliability and validity with a Cronbach’s α coefficient of 0.88 [36]. 

### 2.3. Procedures 

HIV-diagnosed individuals were enrolled when they attended Changsha CDC to get the HIV-infection diagnosis certification or the first CD4 test after diagnosis. The baseline survey was conducted between 1 March, 2013, and 30 September, 2014. One year later, participants who completed baseline survey were contacted for the follow-up survey and all assessments were completed by 31 October, 2015. Participants, at both baseline and one-year follow-up surveys, were asked to complete a questionnaire, including socio-demographic and psychosocial information, via self-report formats. The socio-demographic information was collected at baseline survey and the psychosocial information was collected at both time surveys. Members of our research team reviewed medical records with participants’ permission from Chinese HIV/AIDS Comprehensive Response Information Management System (CRIMS), to obtain the clinical information, including the CD4 counts and HIV-related symptoms of participants at diagnosis and at one-year follow-up. Information on whether participants started ART during the follow-up period was also reviewed from CRIMS at the one-year follow-up survey. The IEC of the Institute of Clinical Pharmacology at Central South University (CTXY-120033-3) approved this study, and written informed consent was obtained from each participant before participation.

### 2.4. Statistical Analysis

Descriptive statistics were presented as frequency with percentages and medians with interquartile range (IQR). The Chi-square and Mann–Whitney U tests were used to compare the baseline sample characteristics between participants who completed two-time surveys and those who dropped out of follow-up survey. The change in scores between baseline and follow-up for depression, anxiety, HIV-related stress, and social support (psychosocial status) were calculated by subtracting baseline scores from follow-up scores. The Mann–Whitney tests were used to compare change scores of psychosocial variables between HIV-positive GBM. 

Two separate generalized linear models (GzLM) were performed separately to determine whether HIV-positive GBM have different trajectories of mental health (depressive and anxiety symptoms) one year after diagnosis, after controlling age, marital status, education, employment, monthly income, living alone or not, having children or not, CD4 counts, HIV-related symptoms, ART status, mental health care utilization, change in stress scores (3 dimensions), and change in social support scores (3 dimensions). All analyses were performed using SPSS for Windows 25.0 (SPSS, Inc., Chicago, IL, USA). A *p* value of <0.05 was considered to be statistically significant.

## 3. Results

### 3.1. Participants

Among the 354 participants who identified their sexual orientation as gay or bisexual, 235 were gay and 119 were bisexual, of which 171 and 87, respectively, continued to complete follow-up survey one year after diagnosis. The participant flowchart is shown in Figure 1. 

There were no significant differences between the 171 HIV-positive gay participants who completed two-time surveys and the 64 who completed only baseline survey, with respect to baseline socio-demographic, clinical, and psychosocial characteristics, except that participants lost to follow-up were more likely to live alone. All baseline sample characteristics were not significantly different between 87 HIV-positive bisexual participants who completed the two-time surveys and 32 of those lost to follow-up (Appendix A).

### 3.2. Participant Characteristics

Table 1 presents the comparison of baseline socio-demographic and clinical characteristics between the gay and bisexual group. Compared to the gay participants, bisexual participants were more likely to be married and have children. Additionally, the bisexual group was more likely to receive ART and access to mental health care than the gay group.

### 3.3. Comparisons of Psychosocial Characteristics Between the Gay and Bisexual Group

Table 2 shows the differences in psychosocial characteristics between the gay and bisexual group. At baseline, there were no statistical differences in psychosocial characteristics between the two groups. While at the one-year follow-up survey, the gay participants had higher PHQ-9 and GAD-7 scores than bisexual, indicating more severe depressive and anxiety symptoms.

### 3.4. Differences in the Trajectories of Psychosocial Status between the Gay and Bisexual Group

For the overall HIV-related stress and social support, the group differences in changes from baseline to follow-up did not reach statistical significance. However, the change scores of PHQ-9 and GAD-7 as well as the social stress were statistically different between the two groups, with smaller decreases in PHQ-9, GAD-9, and social stress score observed in gay participants. The details of changes in psychosocial status between the two groups are shown in Table 3. 

### 3.5. Multivariate Analysis of the Factors Associated with Changes in Depression and Anxiety One Year After Diagnosis

Table 4 shows the multivariate analysis of the factors associated with trajectories of depression and anxiety one year after diagnosis. The difference in changes in depression and anxiety were still statistically significantly between the gay and bisexual group, after controlling all other variables. Compared to bisexual participants, gay participants had smaller decreases in PHQ-9 (*β* = 1.61, *p* = 0.041) and GAD-7 (*β* = 1.54, *p* = 0.027) scores, indicating gay participants have a poor recovery from depressive and anxiety symptoms one year after diagnosis.

In addition, other factors associated with changes in PHQ-9 and GAD-7 scores were mental health services utilization and ART status during the one-year follow-up period, changes in social stress, and objective support scores. Participants who received ART during the first year after diagnosis had better improvement in depressive (*β* = −2.14, *p* = 0.008) and anxiety (*β* = −2.00, *p* = 0.006) symptoms, compared to those who did not. Participants who had access to mental health care after diagnosis were more likely to have better improvement in depressive (*β* = −3.51, *p* = 0.003) and anxiety (*β* = −3.81, *p* < 0.001) symptoms. Increases in social stress scores were associated with increases in PHQ-9 (*β* = 0.43, *p* < 0.001) and GAD-7 (*β* = 0.40, *p* < 0.001) scores, while increases in objective support were associated with decreases in PHQ-9 (*β* = −0.37, *p* < 0.001) and GAD-7 (*β* = −0.36, *p* < 0.001) scores. 

### 3.6. Mental Health Care Utilization Among the Gay and Bisexual Groups

Figure 2 shows the utilization of mental health care among the gay and bisexual groups. Among the 258 gay and bisexual participants (whole sample) who completed both baseline and follow-up surveys, 109 (42.2%) and 81 (31.4%), respectively, reported suffering from significant depressive symptoms (PHQ-9 ≥ 10) and significant anxiety symptoms (GAD7 ≥ 10) at baseline, of which only 23 (21.1%) and 17 (21.3%), respectively, had access to mental health care during the first year after diagnosis. In addition, bisexual participants with significant depression or anxiety, were more likely to assess to mental health care than gay participants (depression: χ^2^ = 5.599, *p* = 0.018; anxiety: χ^2^ = 7.128, *p* = 0.008). 

## 4. Discussion

The depressive and anxiety symptoms experienced by HIV-positive GBM at diagnosis alleviated one year later. These improvements, however, contrary to the hypothesis, tended to be smaller in gay participants. After controlling other variables, sexual orientation was found to be independently associated with the changes in depression and anxiety one year after diagnosis, which further demonstrate the disparities in trajectories of mental health between these two groups. Other factors, including mental health care utilization and ART initiation status during one-year follow-up period, changes in social stress scores, and objective support scores were also shown to be associated with the changes in depression and anxiety, which may provide some reasonable explanations for the disparities in trajectories of mental health between gay and bisexual participants.

Participants who accessed mental health care after diagnosis had larger improvements in depressive and anxiety symptoms in this study, and gay participants with significant depression or anxiety were found to be less likely to use mental health care than their bisexual counterparts. This may be the possible explanation why gay participants had worse improvement in mental health one year after diagnosis than bisexual participants. Furthermore, a low rate of mental health utilization was also found in the whole sample, with less than one-fourth of participants who reported experiencing significant depression and anxiety at diagnosis accessing mental health care after diagnosis. 

It is evidenced that the mental health burden related to HIV/AIDS still remains a major challenge for HIV-positive individuals [37,38,39]. A systematic review has shown that the median prevalence of depression and anxiety among HIV-positive individuals are 60.64% and 43.13% in China, respectively [40]. Unfortunately, poor sustainability and fragmented care of mental health care remains largely an unresolved issue in China [24]. According to the Gelberg–Andersen Behavioral Model for vulnerable populations, the HIV-related stigma might be an important factor associated with the mental health care utilization [41]. The fact that HIV/AIDS remains a highly stigmatized disease is significant in China, which may be a vital obstacle for HIV-positive individuals to access mental health care [42,43]. Furthermore, sexual minority men who seek help may fear of the feelings of shame, embarrassment, and discrimination from healthcare providers if they disclose their true sexual orientation [44]. Consequently, HIV-positive sexual minority men suffer from dual stigma, namely, HIV infection and sexual minority status, which may further prevent them from accessing to mental health care. Compared to gay men, bisexual men may be less likely to disclose their sexual orientation and less visible to others as belonging to sexual minority group [45]. This may also be the reason why bisexual participants were more likely to utilize mental health care than gay participants in this study.

A previous study reported that the lack of professional mental health care specific to sexual minority men and the fear of discrimination from potentially exposing their sexual minority status to healthcare providers, are important potential barriers that prevent sexual minority groups from seeking mental health care [46]. Some sexual minority men may also choose to deliberately hide their sexual orientation when they seek help from mental healthcare providers. If so, this situation may also result in additional negative impacts on their mental health and on the effectiveness and quality of the care they receive from mental healthcare provider [47]. On one hand, before providing care to HIV-positive sexual minority men, mental healthcare providers should achieve professional education and training on issues of sexual minority men and create a trusted medical environment where sexual minority men can disclose and talk about their sexual orientation and related mental health problems associated with HIV/AIDS [48]. On the other hand, in HIV/AIDS care, we should pay more attention to these vulnerable populations (i.e., HIV-positive GBM) who might experience elevated levels of psychological distress but also be afraid to seek mental health care due to fear of internalized stigma or potential external discrimination from healthcare providers. Routine mental health screening and timely referral for patients with severe mental health problems are important in HIV care to improve the mental health outcomes among HIV-positive GBM [49]. 

In this study, we found that the proportion of ART initiation after diagnosis among gay participants was lower than that among bisexual participants, and ART status was a significant factor associated with the changes in depression and anxiety in the GzLM analysis. Participants who received ART during the first year after diagnosis had better improvements in depressive and anxiety symptoms, compared to those who did not. It is widely accepted that early ART initiation is important to the health of HIV-positive individuals [50]. In previous studies, the same results with the positive association between ART initiation and mental health among people living with HIV were also reported [51,52]. Early ART initiation can improve physical health, prolongs life expectancy, and thus improve mental health among HIV-positive individuals [53]. Extensive policy propaganda of ART benefits is important among HIV-positive sexual minority men in China. 

We found that change in social stress were associated with changes in depressive and anxiety symptoms. The lesser the decreases in social stress levels one year after diagnosis, the lesser the improvement in the symptoms of depression and anxiety. The majority of individuals have to struggle with multiple social stress after receiving HIV diagnosis, such as disclosure concerns, HIV-related stigma, and discrimination [54]. These have been previously associated with mental health among people living with HIV [55]. Our study further confirms this association using longitudinal data with two-time points. The different levels of decreases in social stress between HIV-positive GBM has been founded in this study, which may be a possible explanation for the disparities in the trajectories of mental health between these two groups. Compared to gay men, bisexual men may be less visible to others as belonging to a sexual minority and hence may experience less social stigma and discrimination. Additionally, homosexual marriage has not been legalized in China, while gay men may experience greater levels of social stress than bisexual men. It may be the case that compared to bisexual men, gay men who choose heterosexual marriage are more likely to have sex for the purpose of procreation instead of pleasure, and may treat sex in marriage as “work” [56]. It is also difficult for HIV-positive gay men to find a same-sex partner if their HIV-positive status is disclosed [45,57]. Moreover, some studies revealed that, in the gay community, gay men infected with HIV are often considered to be more promiscuous and they should be responsible for their infection [58]. 

Consistent with previous studies showing that a higher level of social support was linked to a better mental health [59,60,61], this study found that increases in the level of objective social support are associated with improvements in depressive and anxiety symptoms experienced by HIV-positive GBM. Social support has been consistently identified as an important moderator that can mitigate the negative psychological consequences of stressful experiences [62]. How to improve the level of available social support should be the focus of future HIV/AIDS research. Our study also found that there was no statistical difference between the gay and bisexual group in the level of social support, at the two-time points, as well as the change in social support, indicating that the social support levels between these two groups may not be a factor that contributes to the disparities in trajectories of mental health between them. Nevertheless, given the important role of social support in the process of coping with HIV diagnosis among newly HIV-infected individuals [60,61], enhancing social support should be considered as an important intervention for HIV-positive sexual minority men to improve the mental health.

We found that compared to gay participants, bisexual participants are more likely to be married and have children. Although these two factors were not significantly associated with the trajectories of mental health in the GzLM analyses, they may be important factors that indirectly influence the changes in mental health among gay and bisexual men over the one year after diagnosis. Previous studies suggested that compared to single men, married men are more likely to start ART after diagnosis, because they need to consider the potential risk for infecting their family members such as wife, children [63,64]. Previous studies also found that social support from spouse or children, could have a positive effect on the ART adherence, such as making medication reminders or increasing confidence to cope with the side effects from ART [65,66]. The potential interaction of these factors (marriage, children, ART initiation) may have a significant influence on the trajectories of mental health among HIV-positive GBM. Future studies should confirm this possibility and explore further potential explanations for the disparities in the trajectories of mental health between HIV-positive GBM.

This study has several limitations. First, the sample in this study is recruited consecutively, which may limit the generalizability of results. Second, we have an unbalanced sample size between the HIV-positive gay and bisexual group, and some participants did not participate in the follow-up study a year later. These participants may exhibit different psychosocial status during follow-up, which may bias the final results. Third, our assessments of depression and anxiety were relied solely on self-reported symptoms scales. Multiple assessments of depression and anxiety, especially the diagnosis tools, would provide a more reliable measure of mental health in this sample. Additionally, in the three dimensions of sexual orientation (i.e., identity, behavior, and attraction), this study only focused on the identity dimension. This approach used in this study may miss some HIV-positive MSM populations who may have distinct mental health needs.

## 5. Conclusions

The depressive and anxiety symptoms experienced by HIV-positive GBM at diagnosis alleviated one year later. These improvements tended to be smaller in gay participants. Other factors including mental health care utilization and ART initiation status during one-year follow-up period; changes in social stress scores and objective support scores were also shown to be associated with the changes in depression and anxiety. Moreover, all these factors, except for change in objective social support, were found to be statistically different between HIV-positive GBM. Special attention should be given to the mental health of HIV-positive gay men. Promoting HIV-positive sexual minority men to assess to mental health services and ART may be important for these populations to improve mental health. Enhancing social support and reducing stress levels may also be necessary for the vulnerable HIV-positive sexual minority groups.

## Figures and Tables

**Figure 1 ijerph-17-03414-f001:**
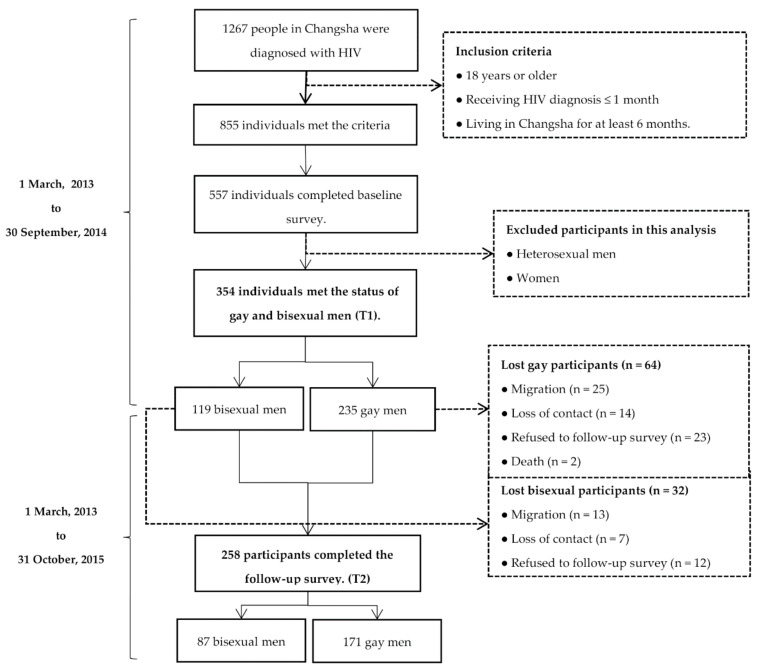
The participants’ flowchart.

**Figure 2 ijerph-17-03414-f002:**
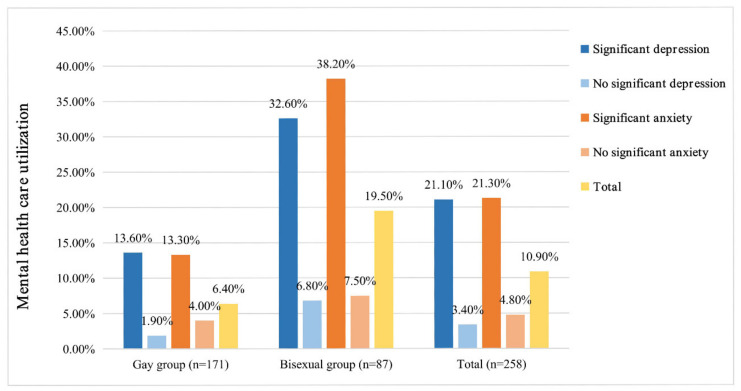
The utilization of mental health care among the gay and bisexual groups.

**Table 1 ijerph-17-03414-t001:** Difference in sample characteristics between the gay and bisexual group ^1^.

Characteristics	Gay Men (*n* = 171)	Bisexual Men (*n* = 87)	*p*
Age			
18–29	115 (67.3%)	60 (69.0%)	0.781
>29	56 (32.7%)	27 (31.0%)	
Marital status			
Single	141 (82.5%)	62 (71.3%)	0.048
Married	18 (10.5%)	19 (21.8%)	
Divorced	12 (7.0%)	6 (6.9%)	
Employment			
Employed	111 (64.9%)	57 (65.5%)	0.565
Unemployed	60 (35.1%)	30 (34.5%)	
Education background			
Senior or lower	70 (40.9%)	38 (43.7%)	0.673
College or higher	101 (59.1%)	49 (56.3%)	
Monthly income (Yuan)			
≤4000	96 (56.1%)	54(62.1%)	0.361
>4000	75 (43.9%)	33 (37.9%)	
Living alone			
Yes	46 (26.9%)	29 (33.3%)	0.282
No	125 (73.1%)	58 (66.7%)	
Children			
With	21 (12.3%)	21 (24.1%)	0.015
Without	150 (87.7%)	66 (75.9%)	
HIV-related symptoms			
With	61 (35.7%)	37 (42.5%)	0.283
Without	110 (64.3%)	50 (57.5%)	
CD4 counts (cells/mm^3^)			
≤350	66 (38.6%)	40 (46.0%)	0.255
>350	105 (61.4%)	47 (54.0%)	
ART initiation status			
Yes	38 (22.2%)	36 (41.4%)	<0.001
No	133 (77.8%)	51 (58.6%)	
Mental health care			
Utilize	11 (6.4%)	17 (19.5%)	<0.001
Not utilize	160 (93.6%)	70 (80.5%)	

^1^ Chi-square tests; ART: antiretroviral therapy.

**Table 2 ijerph-17-03414-t002:** Differences in psychosocial characteristics at two-time points between the gay and bisexual group (median [interquartile range]).

Psychosocial Characteristics	Baseline (*n* = 258)			Follow-Up (*n* = 258)		
	Gay Men (*n* = 171)	Bisexual Men (*n* = 87)	*p*	Gay Men (*n* = 171)	Bisexual Men (*n* = 87)	*p*
Depressive symptoms						
No significant	105 (61.4%)	44 (50.6%)	0.096 ^1^	145 (84.8%)	77 (88.5%)	0.416 ^1^
Significant	66 (38.6%)	43 (49.4%)		26 (15.2%)	10 (11.5%)	
Anxiety symptoms						
No significant	124 (72.5%)	53 (60.9%)	0.058 ^1^	148 (86.5%)	82 (94.3%)	0.060 ^1^
Significant	47 (27.5%)	34 (39.1%)		23 (13.5%)	5 (5.7%)	
PHQ-9	8 (4, 13)	9 (4, 15)	0.111 ^2^	5 (1, 8)	3 (1, 5)	0.021 ^2^
GAD-7	6 (3, 10)	7 (4, 13)	0.101 ^2^	4 (0, 7)	2 (0, 5)	0.021 ^2^
HIV/AIDS-related stress	21 (14, 30)	24 (13, 35)	0.268 ^2^	14 (8, 21)	13 (6, 20)	0.625 ^2^
Social stress	12 (7, 16)	12 (8, 16)	0.394 ^2^	7 (5, 13)	6 (3, 11)	0.097 ^2^
Emotional stress	6 (3, 10)	6 (3, 12)	0.448 ^2^	3 (1, 6)	3 (1, 6)	0.893 ^2^
Instrumental stress	4 (1, 7)	5 (2, 8)	0.191 ^2^	2 (1, 5)	2 (0, 5)	0.906 ^2^
Social support	28 (23, 33)	28 (23, 32)	0.840 ^2^	26 (20, 32)	28 (22, 33)	0.286 ^2^
Subjective support	13 (10, 17)	14 (11, 19)	0.579 ^2^	13 (11, 18)	14 (11, 19)	0.156 ^2^
Objective support	8 (6, 10)	8 (5, 9)	0.432 ^2^	6 (4, 8)	6 (5, 7)	0.750 ^2^
Support utilization	6 (5, 7)	6 (5, 7)	0.896 ^2^	6 (5, 7)	6 (5, 8)	0.455 ^2^

^1^ Chi-square tests; ^2^ Mann–Whitney U tests; PHQ-9: The 9-item Patient Health Questionnaire Depression Scale; GAD-7: The 7-items Generalized Anxiety Disorder Scale.

**Table 3 ijerph-17-03414-t003:** Differences in trajectories of psychosocial status between the gay and bisexual group (median [interquartile range]) ^1^.

Psychosocial Characteristics	Change in Scores		*p*
	Gay Men (*n* = 171)	Bisexual Men (*n* = 87)	
PHQ-9	−2 (−7, 1)	−4 (−10, −1)	0.001
GAD-7	−2 (−7, 1)	−4 (−9, −1)	0.003
HIV/AIDS-related stress	−6 (−15, 1)	−9 (−18, −1)	0.105
Social stress	−3 (−8, 2)	−5 (−9, 0)	0.044
Emotional stress	−2 (−6, 0)	−2 (−7, 0)	0.376
Instrumental stress	−1 (−4, 1)	−1 (−5, 1)	0.270
Social support	−1 (−6, 2)	0 (−5, 4)	0.381
Subjective support	0 (−4, 4)	1 (−4, 5)	0.357
Objective support	−1 (−3, 1)	−1 (−3, 1)	0.812
Support utilization	0 (−1, 1)	0 (−1, 1)	0.446

^1^ Mann–Whitney U tests; PHQ-9: The 9-item Patient Health Questionnaire Depression Scale; GAD-7: The 7-items Generalized Anxiety Disorder Scale.

**Table 4 ijerph-17-03414-t004:** Multivariate analysis of the factors associated with changes in depression and anxiety one year after diagnosis.

Variables	Change in PHQ-9		Change in GAD-7	
	*β* (95% CI)	*p*	*β* (95% CI)	*p*
Sexual orientation				
Bisexual	Ref	0.041	Ref	0.027
Gay	1.61 (0.06, 3.14)		1.54 (0.18, 2.91)	
Mental health care				
Not utilize	Ref	0.003	Ref	<0.001
Utilize	−3.51 (−5.80, −1.23)		-3.81 (−5.85, −1.79)	
ART initiation status				
No	Ref	0.008	Ref	0.006
Yes	−2.14 (−3.71, −0.57)		−2.00 (−3.34, −0.56)	
Change in social stress scores	0.43 (0.32, 0.53)	<0.001	0.40 (0.30, 0.49)	<0.001
Change in objective support scores	−0.37 (−0.57, −0.17)	<0.001	−0.36 (−0.54, −0.18)	<0.001

ART: antiretroviral therapy; Ref: reference group; PHQ-9: The 9-item Patient Health Questionnaire Depression Scale; GAD-7: The 7-items Generalized Anxiety Disorder Scale.

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
