# Peer review of "The Disparities in Mental Health Between Gay and Bisexual Men Following Positive HIV Diagnosis in China: A One-Year Follow-Up Study"

_ijerph, 2020, doi:10.3390/ijerph17103414_

Round 1

Reviewer 1 Report

Thank you for allowing me to review this manuscript. The present study sought to investigate differences and changes in depression and anxiety between gay and bisexual sample one year after HIV diagnosis. Furthermore, the authors sought to identify the factors associated with changes in depression and anxiety and to determine the use of mental health care and differences in that utilization related to sexual orientation.

The study is interesting and well written. The theoretical background presented is adequate. Despite this, some aspects should be better defined and could be improved.

In detail:

Line 15. The authors should spell out ART.

Lines 59 to 71. Aims and hypotheses should be better clarified. The authors should but aims before the hypotheses. Furthermore, hypotheses should not be reported as questions.

Line 101/106. The authors should better clarify why they have tested sexual orientation twice with different measures

Lines 154 to 161. The authors should include the participants, the exclusion criteria, and the procedures in the right section of participants and procedure. Considering the amount of data, it could be useful to split the section of participants from the procedures one. Furthermore, the procedure section should be added after the instruments.

Line 152. In light of the comment above, the authors should modify this part by adding sub-sections.

Line 166. Please insert footnote for all the tables in the text. It could be misleading to report such detailed information on the sample lost, as well as doing analyses on it. The authors should focus only on the information at the baseline of the valid sample.

Line 176. Table 2 reports the same information of the one reported in Table 1, except for the sample lost. Considering this, authors could merge sub-section and comments on results related to Tables 1 and 2 in a unique sub-section.

Line 195. This section could be misleading. The authors should clarify why they talk about associations. Furthermore, they reported results on objective support, although it was not significant in previous tables. Furthermore, the results from line 196 to 203 refer to the whole sample or just the bisexual one? The authors should better clarify this section and the data concerning the gay sample.

Line 207. Please clarify what Model 1 and 2 refer to.

Line 219. The authors should clarify why, although the unbalanced sample, results are more significant in the bisexual sample than in the gay sample. Furthermore, the unbalanced sample should be added as a limitation of the current study.

Reviewer 2 Report

This is a relevant and topical paper addressing differences in gay and bisexual men in mental health within their first year of HIV diagnosis in Changsha, China. The finding that bisexual men tend to show greater improvement in mental health one year after HIV diagnosis is an interesting one, and potentially has practical implications for public health policy and practice.

The comments I have on the paper are primarily related to contextualising the paper for an international readership, more clearly recognising some of the limitations and nuance in interpreting this kind of data, strengthening the conclusion and some comments on clarity of English.

Terminology: This paper is obviously focused on difference between gay and bisexual men. I therefore think it’s important to be clear on the categorisation being used. When participants described themselves as gay or bisexual, were they asked to do this using the English terminology, or using Chinese terminology? If the latter, are there any nuances to the Chinese terms used that might not be immediately apparent in the English translation? I would suggest including the specific identity terms used at least once within the paper.

It might also be helpful to more clearly acknowledge either in the methodology or the limitations that the term ‘MSM’ is often used in the HIV literature because not all men who have sex with men identify as gay or bisexual (or will admit to identifying as such in a survey by health professionals). In the three dimensions of sexual orientation referenced at lines 46-48, this study focused on identity, and not behaviours or attractions. The approach to collecting data in this study will therefore have missed some HIV+ MSM populations (who may themselves have distinct mental health needs).

Relationship networks and support: It appears that within the sample, the majority of participants were not living alone (73% of gay men, 67% of bisexual men). However, the discussion focuses quite heavily on the 20% of bisexual men and 10% of gay men who were married (despite the fact that the models were already adjusted for marital status– suggesting that this should not be a factor in the discrepancy). Given that the discussion and conclusion suggests that social support may be an important area of difference between the two groups, and a route for intervention, it might be appropriate to consider other social networks and support that might be available to HIV+ GBM in Changsha that *don’t* revolve around heterosexual marriage – e.g. same sex partnerships, friendship networks, extended family. If this data is not currently available, perhaps further exploration of HIV+ men’s social and support networks would be a further direction of research and/or intervention.

Conclusion: Overall, this section is a little short and bland. It suggests some potential interventions, but is rather vague on who should provide  them and how. (Who is “We” at line 289? How can support and skills best be provided to this community, or does there need to be further research done to establish this?) The paper at several points correctly references diversity within GBM/MSM populations. It suggests that reasons for distinctions between self-identified gay men and bisexual men might be factors such as stigma, being in a heterosexual marriage, engagement with local LGBT communities, and access to mental health support. Many of the potentially protective behaviours that were more prevalent among bisexual men were, nonetheless, still minority phenomena among bisexual men in the sample (i.e. most bisexual men in the sample weren’t married, didn’t have children, didn’t access mental health support). I am therefore not sure that the conclusion that gay men need targeted, specific support separate from bisexual men is entirely justified – perhaps a better suggestion might be to work with local LGBT+ communities to explore and address the mental health and support needs of HIV positive GBM who do not have access to traditional networks of support such as heterosexual family structures – which this paper suggests may be more common among gay men, but also be an issue for some bisexual men too.

Comments on specific lines of text:

Lines 198-199. Wording is not quite correct on sentences like “Participants who started ART were associated with better improvement in depression and anxiety during the first year”. Should read something like: “Starting ART was associated with better…” Similarly “Participants who utilised…”should start “Utilisation…”

Line 238-239. Insert “people” after LGBT. The sentence “civil union and LGBT organizations are legally allowed and recognized” is slightly confusing – does this mean civil unions are allowed and recognised, or that organisations campaigning for civil unions are allowed? In addition to commenting on the legal freedoms within China, it would be useful to briefly mention broader social attitudes to LGBT+ people - is there stigma or discrimination? Either here or in the methodology section it might be helpful to very briefly provide some more details about Changsha specifically (e.g. is it a large cosmopolitan city? Is there an established network of LGBT organisations?) At line 280, it might also be helpful to more clearly state in what contexts findings from Changsha are likely to be applicable – e.g. perhaps they would be relevant to other urban areas in China but are less likely to be relevant to rural communities?

Lines 252-258. I think what you are trying to say is that bisexual men may be less visible to others as belonging to a sexual minority and hence experience less social stigma and discrimination. However, as it reads, it suggests that being in contact with the LGBT community could itself be a cause of social stress.

Line 264. Use “HIV/AIDS” rather than “AIDS” here?

Lines 267-8. This suggests that the main reason bisexual men are more likely to get married than gay men could be because women are more sexually attracted to bisexual men than to gay men. However, the papers referenced are not about women’s sexual preferences, but about gay/bisexual men’s intentions to marry. On a quick read through, the papers reference a range of factors, such as the men's sexual orientations, filial piety, internalised homophobia, age of realisation of sexuality and family support – but not women’s sexual preferences. Possibly this is a mis-phrasing (perhaps it should read “women are more sexually attractIVE to bisexual men” rather than “sexually attractED”)?

Line 280: “Our results may be limited to all GBM newly diagnosed with HIV” should I think be rephrased to something like “Our results may not be applicable to all GBM newly diagnosed with HIV.”

Reviewer 3 Report

I want to thank you for the opportunity to review this manuscript. The time spent creating and shipping it is greatly appreciated. IMHO, it offers interesting results that can benefit from the scientific community and healthcare professionals. However, currently the manuscript presents some problems that must be taken into account and repaired. Below I present my recommendations separated by sections. Hopefully they will be useful:

Method:

  1. Sample and procedure. How do you deal with initial and follow-up surveys? Please indicate the presentation format.
  2. Instruments. Please note that the expression " by thousands of studies" on the SSRS scale is not appropriate. I imagine that they will not have counted all of these. So I suggest that they change it to "various", "a large number" or another similar expression that determines a more ambiguous quantity.

Results:

  1. Information on the date of data collection and the number of initial and final participants will be more relevant in the "Sample and procedure" section.
  2. A mere curiosity that I would like to know: if the data collection was carried out one year after the diagnosis, that is, in late 2015, why present the data now, 5 years later?
  3. In the tables, please indicate the statistics for each column.

Discussion:

  1. Line 231. Eliminate the percentages.
  2. Line 256-259. Please clarify how you think counting the problem can lead to increased social stress.
  3. In order to better understand the results and discussion of their study, it would be appropriate for the introduction to comment on the situation regarding the legality of gay marriage and the social integration of this group in China.

Thanks for your contribution.

Round 2

Reviewer 1 Report

Dear Authors,

Thank you for having made the suggested revisions. Furthermore, I would like to thank you for the clearness of your explanations and to answer point-by-point to my previous considerations and comments.

Indeed, you answered all my requests.

Best regards.  

Reviewer 3 Report

Thank you for your good work and your diligent responses. Congratulations to the authors for this manuscript.